# Gender-Based Analysis of Oral Health Outcomes Among Elite Athletes

**DOI:** 10.3390/sports13050133

**Published:** 2025-04-27

**Authors:** Alejandro Carlos de la Parte-Serna, Francesca Monticelli, Francisco Pradas, Miguel Lecina, Alejandro García-Giménez

**Affiliations:** 1Faculty of Health and Sports, University of Zaragoza, 22002 Huesca, Spain; acdelaserna@unizar.es (A.C.d.l.P.-S.); fmontice@unizar.es (F.M.); 2ENFYRED Research Group, Faculty of Health and Sports, University of Zaragoza, 22002 Huesca, Spain; franprad@unizar.es (F.P.); alejandro.garcia@unizar.es (A.G.-G.)

**Keywords:** public health, sports dentistry, epidemiology, dental trauma injuries, gender, oral healthcare, sports competition, functional diagnostics

## Abstract

**Background:** Research in dental science reveals a need for enhanced oral healthcare among elite athletes due to the stress generated by excessive exercise. Consideration of the inherent biological distinctions between men and women is crucial in sports dentistry. **Objectives:** Thus, this study aims to analyse the differences in oral health status among elite athletes based on gender. A total of 186 elite athletes (150 men and 36 women) recognised in the region of Aragon (Spain) participated in this study (mean age 24.99 ± 9.34), all of whom practise individual sports. **Methods:** Oral health status analysis included: periodontal, malocclusion and tooth status measured using the restoration index (RI) and the decayed, missing and filled teeth (DMFT) index. **Results:** Women had fewer teeth, a lower restoration index (*p* < 0.05) and lower DMFT index values (*p* < 0.001); furthermore, this group showed a greater number of missing teeth (*p* < 0.001) and decayed teeth (*p* < 0.05). There were no statistical differences in malocclusion, plaque, gingival bleeding, dental erosion or bruxism values between the genders. Mouthguard usage was low (men = 9.4% vs. women = 14.3%; *p* = 0.57). **Conclusions:** This study highlights the need for a multidisciplinary approach to address the high prevalence of oral health issues among elite athletes, despite the differences in health status between men and women.

## 1. Introduction

The health and well-being of professional athletes are essential for injury prevention and optimal performance, ultimately helping to reduce the substantial financial burden associated with sports injuries [1,2]. Research has shown that injuries can lead to significant economic consequences for both individual athletes and sports organisations, while prevention strategies have proven effective and cost-efficient [3,4]. In recent years, athletes have undergone comprehensive medical evaluations conducted by multidisciplinary teams, involving specialists in orthopaedics, nursing, psychology, nutrition and physiotherapy [5]; however, dentistry remains inadequately integrated into these medical teams [6,7].

Prominent organisations such as the European Association for Sports Dentistry (EA4SD), the Academy for Sports Dentistry (ASD) and the FDI World Dental Federation (FDI) have sought to raise awareness among athletes, coaches and medical staff regarding the importance of oral health [8]. These organisations promote orofacial injury prevention and advocate for international policies aimed at advancing research, education and clinical practice in sports dentistry [9]. According to the National Athletic Trainers’ Association (NATA) position statement and the Spanish Society of Sports Dentistry, trainers should be knowledgeable in basic dental care protocols, and medical teams should include a clinician responsible for managing athletes’ oral health (OH) [10].

The primary responsibilities of sports dentistry include managing in-competition dental injuries—such as avulsions, luxation and fractures—and implementing preventive measures like mouthguards. Mouthguards are particularly critical in high-risk sports (e.g., rugby, boxing, ice hockey and lacrosse), where they significantly reduce the risk and severity of orofacial trauma (OFT). In contact sports, studies have shown that dental injuries occur in 59.5% of athletes not using mouthguards, compared with only 7.5% among users, indicating that non-users are more than twice as likely to suffer OFT [11].

Beyond trauma management, sports dentistry is expanding its scope to address the broader OH risks faced by elite athletes. Participation in high-level sports is associated with an increased risk of oral diseases, which can negatively affect both well-being and athletic performance. Contributing factors include high-carbohydrate diets, frequent consumption of acidic sports drinks and reduced saliva flow during intense exercise—all of which increase the risk of dental erosion, caries and periodontal disease. To mitigate these risks, OH screenings should be integrated into pre-season medical evaluations, and sports dentists should provide tailored, sport-specific OH advice. Promoting the engagement of athletes in their own oral care is essential, as is fostering collaboration among dental professionals with expertise in sports to ensure evidence-based treatment and continued research [12].

Recent studies have highlighted the significant OH challenges faced by athletes [13]. Ashley et al. have reported that 32% of elite athletes experienced oral health-related effects on their performance, including pain, difficulty participating in training and reduced output [6]. These issues were primarily attributed to conditions such as caries, erosive tooth wear and periodontal disease, particularly among team sports athletes. The study emphasised the need for regular oral screenings and targeted health promotion to diminish the performance effects of poor oral health. These findings reinforce the connection between OH and athletic performance and underscore the need for incorporating dental care as a component of comprehensive athlete healthcare.

Multiple studies have documented OH deficiencies in athletic populations [14,15]. Bramantoro et al. [16] found that conditions such as periodontal disease, malocclusion and periapical inflammation were associated with decreased muscle strength, impaired body balance and reduced cardiorespiratory function. Occlusal issues, such as mandibular deflection, have been shown to reduce muscular power by up to 17.7% in elite rowers, demonstrating the direct influence of oral status (OS) on athletic performance. The demand for dental care is particularly evident during major international sporting events. For instance, at the Athens 2004 Olympics, dental care was the second most requested medical service. Nearly 1600 dental treatments were provided during the Beijing 2008 Games, while 30% of all athlete medical emergencies at the London 2012 Olympics were of dental origin [17,18]. A study on Dutch elite athletes prior to the Rio 2016 Olympics revealed that nearly 43% required immediate dental treatment, despite 81% having visited a dentist in the past year [19].

A recent comprehensive review by Schulze et al. confirmed a high prevalence of oral diseases among elite athletes and highlighted the significant roles of dietary habits and oral hygiene practices [20]. The study reported that dental caries prevalence ranged from 20% and 84%; dental erosion from 42% to 59%; gingivitis from 58% to 77%; and periodontal disease from 15% to 41%. These conditions were associated with the frequent consumption of sugary sports drinks, carbohydrate-rich diets and inadequate oral hygiene. Notably, dental pain was cited as contributing up to 18% of performance loss in some athletes.

In the general population, various studies have shown that men tend to have poorer OS and lower engagement with dental care than women [21,22]. Research consistently shows that women possess greater knowledge of oral hygiene practices, demonstrate more interest in OH and have a more positive perception of their own oral condition [21,23]. Women also engage more frequently in good oral hygiene behaviours, such as regular brushing, flossing and dental visits. However, some studies have shown more caries or decayed teeth in women, and this fact could be explained by hormonal differences, pregnancy or differences in diet and saliva pH [24].

Given these documented gender differences, we hypothesise that female athletes maintain better OS and experience lower rates of oral disease than their male counterparts. This study aims to assess key OH indicators—including the decayed, missing and filled teeth (DMFT) index, the restoration index (RI), and the prevalence of malocclusion and oral hygiene behaviours—to explore potential gender disparities in the OH status of elite athletes.

## 2. Materials and Methods

### 2.1. Participants

A total of 186 elite athletes from the Aragon region of Spain joined the study, officially designated as high-performance athletes according to the regional government legislation (Orden ECD 1630/2016) [25]. The sample included 150 men and 36 women, with a mean age of 24.99 ± 9.34 years. Among them, 74 athletes (53 men and 21 women) were engaged in endurance sports or individual sports, while 112 (97 men and 15 women) competed in team sports. The sample size was determined using G*Power 3, targeting a 95% confidence level and a 6% margin of error relative to the total population of elite athletes in Aragon.

All participants received detailed information about the study’s purpose and objectives. They voluntarily agreed to take part and provided informed consent in writing. Each participant was assigned a unique code to ensure anonymity in data handling. The study adhered to the ethical principles set out in the Declaration of Helsinki, last revised at the 2013 World Medical Assembly in Fortaleza, Brazil). Ethical approval was obtained from the Clinical Research Ethics Committee of the Department of Health and Consumption of the Government of Aragon (DGA) (approved code 11/2015).

Inclusion criteria were as follows: age between 18 and 35 years; official high-performance athlete status conferred by the DGA; absence of chronic illness; and a minimum of 5 h of weekly training. Athletes were excluded if they were pregnant, breastfeeding, had not trained in the previous 31 days or failed to complete all stages of the study protocol.

### 2.2. Procedures

All oral assessments were conducted by a single examiner; namely, a recently licensed dentist, who underwent prior training to ensure consistency and reliability in examination techniques and data recording. The training was supervised by a senior dentist with over 10 years of clinical experience. The calibration process involved observing two rounds of oral exams on a group of 10 individuals, with a minimum 30 min interval between evaluations to assess diagnostic repeatability. Intra-examiner reliability was assessed using percentage agreement (matching diagnoses/total diagnoses × 100), with variability kept under 3%, indicating satisfactory diagnostic consistency. OS assessments followed the World Health Organisation’s (WHO) standardised protocol for adults (Version 2013) [26], and all examinations were performed in a controlled clinical environment compliant with WHO standards in terms of lighting, equipment and sanitation.

The OS of each athlete was determined using the DMFT index to identify cases of dental decay, tooth loss or restorative work. The restoration index (RI) was calculated by dividing the number of filled teeth by the DMFT score, multiplied by 100 [27]. Malocclusion was evaluated based on Angle’s classification [28], considering the documented impact of dental occlusion on postural control and athletic performance, especially in precision sports like shooting, golf and running [29,30,31].

Additional assessments included recording dental fluorosis using Dean’s Index; enamel defects using the modified Developmental Defects of Enamel (DDE) Index; and erosion classified by severity. Periodontal status was evaluated using the modified Community Periodontal Index (CPI) [32]. Athletes were also asked about oral habits such as bruxism (teeth grinding), mouth breathing, prior extractions, dental trauma (e.g., fractures) and the timing of their most recent dental visits. To assess subjective perceptions of their oral health, athletes rated their satisfaction on a scale of 1 (completely dissatisfied) to 10 (completely satisfied).

### 2.3. Statistical Analysis

Data were analysed using IBM^®^ SPSS^®^ Statistics version 25.0 for macOS (IBM^®^ Corp., Armonk, NY, USA). The Kolmogorov–Smirnov test was employed to examine the normality of data distribution, and the Levene test was used to assess the homogeneity of variances. Quantitative variables were compared using independent-sample Student’s t-tests. Categorical variables were evaluated using chi-squared tests with Yates’ correction where appropriate. Results were considered statistically significant at *p* < 0.05. Quantitative data are presented as mean ± standard deviation, while categorical variables are expressed as percentages.

## 3. Results

Table 1 shows the characteristics of the subjects included in the study.

Table 2 displays data on the total number of teeth observed, along with the distribution of healthy, decayed, missing and restored teeth. Additionally, the table includes values for the DMFT index and the RI. Male athletes presented more filled teeth than their female counterparts. Both the DMFT and RI scores were significantly elevated in male participants (*p* < 0.05).

Table 3 represents malocclusion where statistical differences were found between both genders in the three classes.

Table 4 outlines the condition of gingival tissues and enamel among the athletes. No statistically significant differences were observed between male and female participants in these variables.

## 4. Discussion

This study aimed to assess OH in elite athletes, with a specific focus on gender-based differences. We hypothesise that female athletes would exhibit better oral conditions and, consequently, experience fewer orofacial diseases and injuries than their male counterparts. While previous research has linked gender to OH, most studies have examined the general population [21], adolescents [23] or differences across sporting disciplines [33]. To the best of our knowledge, this is the first study specifically evaluating the OH of elite athletes from a gender perspective.

We divided the discussion into four sections: (1) a summary and comparison of key findings with the literature; (2) implications for athletic performance; (3) clinical implications and recommendations; and (4) study limitations and future directions.

### 4.1. Summary and Comparison of Key Findings

Our evaluation of tooth status involved four categories: healthy, decayed, absent and filled. Female athletes showed more healthy teeth, consistent with findings from previous research [33,34]. In terms of the total number of teeth, both male and female athletes demonstrated values comparable to previous studies (male athletes: 28.62 ± 2.11; female athletes: 28.38 ± 1.32), aligning closely with data from Needleman et al., who reported an average of 29.7 teeth among Olympic athletes [10].

The number of decayed teeth in our sample was lower than the 3.5 affected teeth reported in early studies on Olympic athletes [35]. Significant gender differences were observed in the DMFT and RI indices, with women demonstrating better outcomes: RI (28.52 ± 24.09 vs. 16.94 ± 23.67; *p* = 0.001) and DMFT (8.66 ± 3.66; *p* < 0.001). Shaharuddin et al. reported similar values in a study involving 84 national-level athletes [36]. Compared with the Spanish young adult population (DMFT = 7.40), our male athletes showed high values, while female athletes were slightly below this average. This may suggest a lack of sufficient dental care among female athletes, despite their better oral status. Oral care appears to be neglected by all genders, as the time since the last dental visit (men: 19.87 ± 13.09 months; women: 17.38 ± 14.59 months) far exceeds the six-month check-up interval recommended by dental health authorities [13,37].

In our sample, 71.6% of participants were men; this male predominance mirrors findings from previous studies, such as Athens 2004 [18] and London 2012 [10], where male participation rates were 72.45% and 57%, respectively. The higher number of treated teeth in male athletes could reflect a historical research and clinical focus on non-dental health concerns in female athletes (e.g., the female athlete’s triad or reproductive health) [38,39,40].

Despite differences in tooth status, no significant gender-related variations were found in periodontal health, gingival bleeding, plaque levels or dental erosion. Similarly, no statistically significant differences in malocclusion prevalence were observed, although some minor variations in Class II and III malocclusions were noted. Mouthguard usage was low across all genders, likely reflecting the types of sports represented in our sample. Overall, our findings suggest that female athletes exhibit better oral health outcomes, although the gender gap may be narrower than in the general population, possibly because of shared environmental and lifestyle factors among elite athletes. Given these results, the socioeconomic factors common to all high-performance athletes included in this type of program (level of education, economic status and assistance benefits) are more relevant than gender [41]. In a retrospective study, Leiva Arcas et al. (2021) found that the majority of athletes included in the Spanish Olympic team from 2008 to 2016 were middle class according to their incomes [42]. In a quantitative, analytical, cross-sectional study, De Lucena et al. (2021) analysed different socioeconomic factors and found that gender was less important than education, age, or economic status [43].

### 4.2. Implications for Athletic Performance

Oral health issues—particularly untreated caries and periodontal disease—can negatively affect physical performance [6,13,14]. As Bramantoro et al. [16] reported, periodontal disease can reduce physical fitness, and our findings support this concern. Dental pain stemming from these conditions may hinder training and competition performance.

In endurance sports requiring in-race nutrition, malocclusions may impair chewing efficiency, leading to gastrointestinal discomfort and nutrient malabsorption, ultimately reducing the energy available for performance [20,44,45]. A cohort study by Lăzureanu et al., [46] involving 155 healthy individuals (78 men and 77 women, aged 30–92 years) found a strong link between periodontal disease and cardiovascular conditions (*p* = 0.001), with no observed gender differences.

Although our study did not find gender differences in malocclusion prevalence, prior research indicates that occlusal issues can impact posture and precision in sports like shooting and gymnastics [47]. Mielle et al. [48] also noted associations between perceived oral health and caries experience among professional athletes. We should investigate these areas further, especially in sports mandating mouthguard use. Notably, malocclusion may impair the fit and protective function of mouthguards. However, sport type appears to be a more relevant factor than gender in determining malocclusion risk [49].

### 4.3. Clinical Implications and Recommendations

Our results support existing recommendations from major sports dentistry organisations, such as the EA4SD, ASD and WHO [34,37,48]. The FDI urges athletes to strengthen their oral hygiene practices, especially given their frequent intake of acidic or sugary energy products that lower oral pH [8]. As women naturally have lower oral pH levels [50], they may require extra attention to hygiene after consuming these substances.

Although female athletes may show more attention to oral care, adherence to oral health guidelines was suboptimal in both groups. Based on the high prevalence of dental concerns, regular dental evaluations should be incorporated into athletes’ annual health check-ups. Sports organisations should consider employing sports dentistry professionals to provide targeted care related to trauma, diet and hygiene.

Despite low reported use, mouthguards are critical for injury prevention in contact sports. Educational efforts should highlight their importance, particularly in high-risk disciplines.

The consumption of high-sugar and acidic products is a known risk factor for oral diseases in athletes. Comprehensive education programs on oral hygiene, hydration, and nutrition should be implemented to reduce these risks.

### 4.4. Limitations and Future Research

Although statistically robust, our sample is geographically limited to elite athletes in Aragon, Spain. Future research should include athletes from other regions and a wider variety of sports to improve generalisability. The relatively small number of female athletes also limits the strength of gender comparisons.

The disparity between men and women in our sample complicates our data analysis; nonetheless, this gender proportion reflects the reality of women in professional sports in 2016.

Although no gender differences in malocclusion were observed, future studies should explore the effects of specific occlusal patterns on performance, particularly in sports requiring fine motor control and postural stability. Longitudinal studies examining oral health across athletic careers could also offer valuable insights into how OH status evolves and influences both performance and recovery over time.

## 5. Conclusions

This study highlighted the need to integrate oral health care into the routine medical management of elite athletes. The findings revealed gender-based differences in oral health status, which may influence athletic performance, underscoring the importance of implementing targeted oral health strategies tailored to the needs of athletes. Given the crucial role that oral health plays in sustaining optimal physical performance, further research is essential to develop comprehensive, sport-specific interventions to address the oral health-related challenges faced by this population. 

## Figures and Tables

**Table 1 sports-13-00133-t001:** Participants’ characteristics.

	Men	Women
N	150	36
Age (years)	25.41 ± 7.20	21.92 ± 4.98
Gender (%)	80.60%	19.40%
Dental self-evaluation (points)	7.96 ± 1.24	7.95 ± 1.60
Last dental check-up (months)	19.87 ± 13.09	17.38 ± 14.59

**Table 2 sports-13-00133-t002:** Number of total teeth and status, DMFT and IR indices.

	Men	Women	*t*-Value	*p-*Value
Total teeth number *(n)*	28.62 ± 2.11	28.38 ± 1.32	3.851	0.001 **
Healthy teeth (*n*)	23.38 ± 3.63	25.05 ± 2.80	−1.652	0.048
Decayed teeth (*n*)	2.30 ± 2.00	1.86 ± 2.10	2.045	<0.001 **
Missing teeth *(n)*	3.45 ± 2.06	3.62 ± 1.32	3.458	0.267
Filled teeth (*n*)	2.99 ± 2.00	1.98 ± 1.00	−0.389	0.001 **
DMFT	8.66 ± 3.66	6.76 ± 2.95	4.586	<0.001 **
RI	28.52 ± 24.09	16.94 ± 23.67	−3.026	0.001 **

DMFT: decayed, missing and filled teeth index; RI: restoration index; ** *p*-value < 0.05.

**Table 3 sports-13-00133-t003:** Prevalence of malocclusion according to gender.

	Men	Women	X^2^	*p-*Value
Class I (%)	41.5	33.3	0.100	0.009 **
Class II (%)	37.7	61.9	0.100	0.009 **
Class III (%)	20.8	4.8	0.100	0.009 **

X^2^: chi-squared value. ** *p*-value < 0.05.

**Table 4 sports-13-00133-t004:** Periodontal pockets, gingival, bleeding plaque, dental erosion and bruxism according to gender.

		Men	Women	*X^2^*	*p-*Value
Periodontal pockets (%)	No	75.5	85.7	0.581	0.726
	Yes	24.5	14.3		
Gingival bleeding (%)	No	11.3	19.0	1.589	0.572
	Yes	88.7	81.0		
Plaque (%)	No	17.0	23.8	3.827	0.098
	Yes	83.0	76.2		
Dental erosion (%)	No	28.3	38.1	3.859	0.168
	Yes	71.7	61.9		
Bruxism (%)	No	30.2	38.1	1.084	0.589
	Yes	69.8	61.9		
Mouthguard ** (%)	No	90.6	85.7	0.348	0.570
	Yes	9.4	14.3		

*X*^2^: chi-squared value. ** neither required in sports assessed nor compulsory, *p*-value < 0.05.

## Data Availability

The data presented in this study are available upon request from the corresponding author. The data are not publicly available due to privacy.

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
