# Peer review of "Gender-Based Analysis of Oral Health Outcomes Among Elite Athletes"

_sports, 2025, doi:10.3390/sports13050133_

Round 1
Reviewer 1 Report
Comments and Suggestions for Authors
Dear Author
You have written an interesting paper focusing on the Oral Health Status of Elite Athletes divided by gender. However, several parts need to be addressed for the manuscript to be clearer.
Abstract:
The main aim-rationale is not clear. Rewrite the satrting section of the abstract.
Introduction:
The introduction effectively presents the relevance of OH in sports linking it to performance and injury prevention. The relevant literature is used.
The authors nicely Identify gender as a variable that has not been fully explored in elite athletes—providing a clear rationale. However, this could be better written and presented.
Methods:
Ethical approval, consent, and inclusion/exclusion criteria are clearly stated and appropriate.
How do you determine elite athletes? Be specific and report the criteria.
How did you calculate the sample size? G*Power or any other software? report
So, what is the total population of elite athletes in this selected region - report?
You have a great disparity between male and female athletes, which is your main focus—hypothesis. 150 VS 36??? This is a very poor methodological background for your hypothesis. Also, I see from the ethics approval that this is a very old dataset—2015. So, from what year is this dataset?
You stated: The sample represents a confidence level=95% with a margin of error=6% /of what? to what do you relate this to?
Why the age limit of 18? You can have elite athletes under the age of 18?
line 128 to be recognized as an elite athlete - by who? according to what classification -R&R
(f) to present good general health - who evaluated this, or was it self-reported?
Why was pregnancy listed as an exclusion factor? this is and odd decision? please elaborate on this, as pregnant females compete in elite competitions and win Olympic medals.
Why only one examiner? a huge limitation -
So the examiner was not a dentist but his supervisor was????
For agreement between raters, ICC or Kendal's coefficient of concordance should be calculated. This is not a standard practice!
At what time of day was the examination done?
Report effect size in the statistics and results.
The limitations section should state the very low number of females in the study that should focus on gender differences.
Several references are not cited correctly in the final references list! Amed them.
Overall a promising study that needs major revision.
Comments on the Quality of English LanguageThe paper needs English proofreading.
Author Response
Response to Reviewer 1 Comments
Dear Reviewer 1,
Thank you for allowing us to submit a revised draft of our manuscript, "Oral Health Status in Elite Athletes: A Gender Perspective," to the issue "Competition and Sports Training: A Challenge for Public Health."
We appreciate the time and the effort that you have dedicated to providing your valuable feedback on our manuscript. As a result, we have been able to incorporate changes to reflect most of the comments provided by you. We have highlighted the changes within the manuscript.
Here is a point-by-point response to your key notes and concerns. We added all these changes to the document, highlighting the corrections in red for easy identification, also added some comments. We hope you find it helpful.
Abstract
We totally rewrote the first paragraph to clarify the study's aim and the background of oral health status.
Methods
- How do you determine elite athletes? Be specific and report the criteria.
Dear reviewer. We strictly followed the official legislation in our country (Spain) and particularly in Aragon. Here can follow the link where it is available and you can consult it. We have added it as a refence in the manuscript (line 109)
Orden ECD 1630/2016 https://deporte.aragon.es/deporte-de-alto-rendimiento/informacion-sobre-la-calificacion-d-a-a-r/tipologias-de-calificacion/id/1345.
Basically, they must have competed in international tournaments or being in high performance centres
- How did you calculate the sample size? G*Power or any other software? Report.
We used G*Power, and we added some extra lines describing the sample size of our study in the manuscript (lines 115-118)
- So, what is the total population of elite athletes in this selected region - report?
In 2015, there were 3893 elite athletes in Spain registered in CSD
https://www.csd.gob.es/es/alta-competicion/deporte-de-alto-nivel-y-alto-rendimiento/deportistas-de-alto-nivel-y-alto-rendimiento/deportistas-de-alto-4
- You have a great disparity between male and female athletes, which is your main focus—hypothesis. 150 VS 36??? This is a very poor methodological background for your hypothesis. Also, I see from the ethics approval that this is a very old dataset—2015. So, from what year is this dataset?
The study reflects the reality of the practice of high-performance sport in Spain with a higher proportion of male than female athletes. Although these figures are now equal, the 2015 study showed that proportion.
Our dataset is from 2016. While we had intended to publish these findings earlier, our research group experienced funding delays that impeded data analysis and publication.
- Why the age limit of 18? You can have elite athletes under the age of 18?
Yes, but we limited our population to adults for two main reasons: a) legal concerns and b) adults are totally responsible for their oral health status whereas teenagers depend on parents' decisions to care their oral health.
- line 128 to be recognized as an elite athlete - by who? according to what classification -R&R
We followed the official legislation in our country
Orden ECD 1630/2016 https://deporte.aragon.es/deporte-de-alto-rendimiento/informacion-sobre-la-calificacion-d-a-a-r/tipologias-de-calificacion/id/1345.
Basically, they must have competed in international tournaments or being in high performance centres
- to present good general health - who evaluated this, or was it self-reported?
Subjects´ health was assessed according to his medical record filed in the Government of Aragon. Athletes granted their permission to access to these data.
- Why was pregnancy listed as an exclusion factor? this is and odd decision? please elaborate on this, as pregnant females compete in elite competitions and win Olympic medals.
Why only one examiner? a huge limitation –
We totally agree, but unfortunately, we had a limited budget. To improve the general quality, we added a senior dentist to check the quality of the check-ups carried out by the junior dentist.
- So the examiner was not a dentist but his supervisor was????
Thank you for your comment. To clarify, the examiner in this study was a recently licensed dentist who received specific training for the oral examinations. While the examiner had recently obtained their license, a senior dentist supervised throughout the process them with over 10 years of professional experience. This ensured the senior dentist carried out the oral examinations to the highest standards. We added it to 'Procedures' (line 130).
- For agreement between raters, ICC or Kendal's coefficient of concordance should be calculated. This is not a standard practice!
Thank you for your valuable comment regarding the assessment of agreement between raters. In our study, only a single examiner conducted all oral health evaluations, following a calibration process under the supervision of a senior dental specialist. As such, inter-rater agreement was not applicable in this case. Instead, intra-rater reliability was evaluated using the simple agreement percentage between repeated assessments, which showed an acceptable consistency with less than 3% variability.
We acknowledge that while Intra-class Correlation Coefficient (ICC) or Kendall's coefficient of concordance are robust methods for assessing agreement they are typically applied in studies involving multiple raters. Given the design of our study and the involvement of only one examiner, we considered simple percentage agreement a suitable and transparent method for confirming diagnostic consistency.
Nonetheless, we appreciate the suggestion and have clarified this point in the revised manuscript
- At what time of day was the examination done?
All examinations were carried out in the morning
- The limitations section should state the very low number of females in the study that should focus on gender differences.
Thank you for your commentary. We have added the ratio of men vs women in the limitations.
- Several references are not cited correctly in the final references list! Amed them.
Thanks for your appreciation, we have added new references and have amended the missing ones
Yours faithfully,
Miguel Lecina
University of Zaragoza

Reviewer 2 Report
Comments and Suggestions for Authors
Thank you for the article, but I expected to see a comparison between people that are not athletes and athletes , or maybe comparing data on oral health before and after starting a professional sport...As it is, it is not very interesting. The findings support existing literature suggesting oral health affects athletic performance, yet the study’s limited generalizability, male-dominant sample, and focus on individual sports narrow its broader applicability. After all, causal link between oral health and actual sports outcomes remains largely speculative, though there are mechanisms that can explain this. The paper reinforces the need to introduce dental professionals within sports medicine teams, but could benefit from stronger interdisciplinary discussion and exploration of socioeconomic factors influencing care. Please, at least address the problems from my last phrase, in order to improve somewhat the quality of the article.
Author Response
Dear reviewer 2,
Thank you for allowing us to submit a revised draft of our manuscript, "Oral Health Status in Elite Athletes: A Gender Perspective," to the issue "Competition and Sports Training: A Challenge for Public Health."
We appreciate the time and effort that you have dedicated to providing your valuable feedback on our manuscript. Consequently, we have been able to incorporate changes to reflect most of the comments provided by you. We have highlighted the changes within the manuscript.
Here is a point-by-point response to your key notes and concerns. All these changes have been added to the main document, and we have highlighted the corrections in yellow colour for you to find them easily. Additionally, some comments have been inserted. We hope you find it helpful.
1- The paper reinforces the need to introduce dental professionals within sports medicine teams but could benefit from stronger interdisciplinary discussion and exploration of socioeconomic factors influencing care.
We thank you for your feedback and have incorporated your suggestions by adding two supplemental paragraphs to the discussion. The evidence of the oral health status comparing men and women is scarce in general population and minimum in athletes. That is the reason because we have focus mainly in the studies including athletes and not in common population. These individuals have worse oral health because two main reasons: first the extreme exercise they realise imply high stress for their oral cavity (use of high-sugar drink, high carbohydrates diets, Ph alterations due to the exercise, etc) and secondly, these athletes focus on other medical care (physicians, nutritionists, orthopaedics, coaches, etc) and rarely pay attention to these issues.
We hope you may find convenient the information added in this email, and please do not hesitate to contact us regarding any queries you might have.
Yours faithfully,
Miguel Lecina
University of Zaragoza

Round 2
Reviewer 1 Report
Comments and Suggestions for Authors
Dear Authors,
Thank you for addressing all of my comments and suggestions. The manuscript quality improved, and I believe the paper should be accepted in its current condition.
Kind regards
Reviewer 2 Report
Comments and Suggestions for Authors
Thank you for taking into consideration my comments.